# ANAH-v2: Scaling Analytical Hallucination Annotation of Large Language Models

Yuzhe Gu[1,2*]   Ziwei Ji[2,3*]   Wenwei Zhang[2†]   Chengqi Lyu[2]   Dahua Lin[2,4,5]   Kai Chen[2†]

[1]Shanghai Jiao Tong University   [2]Shanghai AI Laboratory
[3]Hong Kong University of Science and Technology
[4]MMLab, The Chinese University of Hong Kong   [5]HKGAI under InnoHK
{guyuzhe,zhangwenwei,lvchengqi,chenkai}@pjlab.org.cn   zjiad@connect.ust.hk

## Abstract

Large language models (LLMs) exhibit hallucinations in long-form question-answering tasks across various domains and wide applications. Current hallucination detection and mitigation datasets are limited in domain and size, which struggle to scale due to prohibitive labor costs and insufficient reliability of existing hallucination annotators. To facilitate the scalable oversight of LLM hallucinations, this paper introduces an iterative self-training framework that simultaneously and progressively scales up the annotation dataset and improves the accuracy of the annotator. Based on the Expectation Maximization algorithm, in each iteration, the framework first applies an automatic hallucination annotation pipeline for a scaled dataset and then trains a more accurate annotator on the dataset. This new annotator is adopted in the annotation pipeline for the next iteration. Extensive experimental results demonstrate that the finally obtained hallucination annotator with only 7B parameters surpasses GPT-4 and obtains new state-of-the-art hallucination detection results on HaluEval and HalluQA by zero-shot inference. Such an annotator can not only evaluate the hallucination levels of various LLMs on the large-scale dataset but also help to mitigate the hallucination of LLMs generations, with the Natural Language Inference metric increasing from 25% to 37% on HaluEval. [1]

## 1 Introduction

Large Language Models (LLMs) have shown remarkable capabilities in various tasks [10, 11, 35, 51, 57]. However, they tend to produce *hallucination*, *i.e.*, plausible-sounding but unfaithful or nonsensical information [5, 30], that significantly hinders their real-world applications. Initial steps to address this issue involve the creation of datasets that can help to detect, annotate, and mitigate hallucinations [14, 29, 40]. Since the potential hallucinations of LLMs are in various fields, the spectrum of knowledge in the dataset is expected to be large-scale and comprehensive, covering various domains. Consequently, the size and diversity of datasets are critical for the oversight of LLM hallucinations.

However, constructing and scaling-up hallucination annotation datasets face significant hurdles [8, 9, 29, 43]. One primary challenge is the prohibitively high costs and labor intensity required for their accurate assessment [43, 47], since the fine-grained hallucination annotation requires intensives labor for reading long documents and annotating the hallucination details sentence by sentence. Moreover, due to the insufficiency of accurate human annotations, the reliability of existing hallucination annotators and detectors becomes another pressing concern [29]. These tools have been found to

---

*Equal contribution † Corresponding author
[1]Dataset, code, and model are released at https://github.com/open-compass/ANAH.

produce inaccurate results [9, 46, 61], *e.g.*, even GPT4 [1], one of the most powerful LLMs, is not satisfactory and cannot achieve a compatible performance of humans [29].

Existing works [3, 25, 36, 38, 56, 58, 69] have explored strategies in data augmentation and self-training to extend dataset size and boost the performance of models in the fields of image segmentation, multi-lingual translation, math reasoning, *etc*. However, how to scale the hallucination annotation datasets efficiently is under-explored in the community, which significantly hinders the in-depth analysis and further mitigation of LLMs hallucinations at a large scale.

To address the research gap, this paper proposes an iterative self-training framework designed to scale up the hallucination annotation dataset and simultaneously increase the accuracy of annotators (Fig. 1). The iterative framework can be explained from the perspective of the Expectation Maximization (EM) algorithm. In the Expectation (E) step, we apply the existing best hallucination annotator to estimate the ground-truth hallucination annotations of the scaled dataset. We adopt an inference pipeline on top of the annotator with self-consistency strategy [63] to provide a more robust estimation of the annotations, which lays the groundwork for training a more precise annotator in the subsequent step. In the Maximization (M) step, we combine the existing annotations with the scaled data annotations derived from the previous E steps to train a new hallucination annotator. Training on more data leads to a more accurate annotator and a more robust annotation pipeline, setting the stage for the subsequent round of annotations.

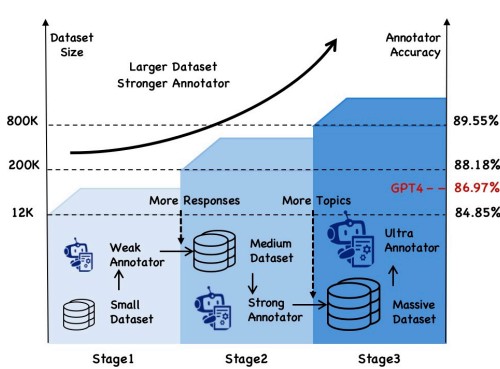

Figure 1: Our iterative self-training framework progressively scales up the hallucination annotation dataset size (left) and simultaneously increases the annotator's accuracy (right) in three stages.

The iterative process consists of three stages of multi-dimensional data scaling as shown in Fig. 1. Initially, we train a weak annotator on human annotations. In the second stage, we collect the hallucination responses from more open-source LLMs for the same questions in the dataset to improve the generalization ability of hallucination annotators to model responses. In the third stage, we expand the number of topics and questions in the dataset and collect hallucination annotations with the more robust annotator. This progressive scaling strategy stabilizes the annotator's performance when evaluating familiar and unfamiliar responses across diverse topics.

Extensive experimental results show that our enhanced annotator significantly outperforms existing models, including the advanced GPT-4, in terms of accuracy. Our annotator not only performs best on the in-domain fine-grained hallucination annotation dataset ANAH (89.24%) but also obtains new state-of-the-art (SOTA) results on HaluEval (81.54%) and HalluQA (94.44%) under zero-shot setting. In addition, the annotator automates the hallucination evaluation on the dataset, offering a comprehensive benchmark for the research community to evaluate the hallucination levels of numerous open-source models, providing a practical reference for future hallucination mitigation of LLMs. Using a simple reranking strategy with the annotator, we reduce the hallucination of the final LLM generations on HaluEval, with the NLI metric increasing from 25% to 37%.

## 2 Related Work

**Self-improvement of Large Models.** As Large Language Models (LLMs) become more and more powerful, the community starts to explore different strategies to achieve the self-improvement of LLMs, *i.e.*, to improve the LLMs using the supervision from LLMs [12, 24, 28, 50, 53]. For example, existing works have explored self-alignment using LLMs with ethical principles [3, 58, 69]. There are also methods [13, 25, 38, 56, 70] strengthen LLM's capabilities on tasks such as reasoning by training the LLMs on the high-quality responses from themselves on the same questions. In the field of computer vision, SAM [36] introduces manual and model-assisted labeling to expand the image segmentation dataset and enhance the performance of image segmentation models. However, the application of self-improvement is under-explored in fine-grained hallucination annotation. This field

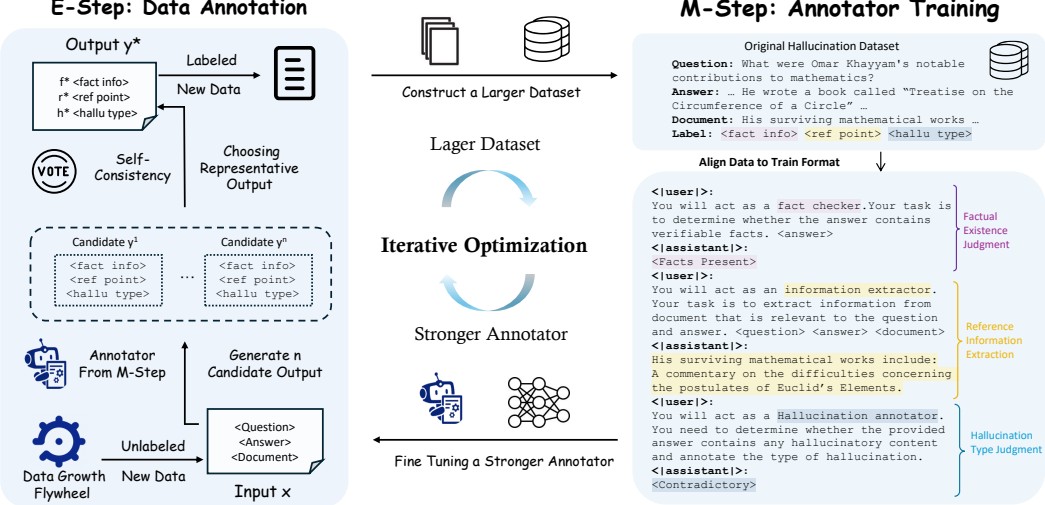

Figure 2: The schema of EM-based interactive self-training framework. In the **E-step**, given unlabeled new data from the Data Growth Flywheel, the annotator predicts N candidate outputs $y$. Then the representative annotation $y^*$ is chosen via *self-consistency*. As a result, we construct a **larger dataset** by collecting the new annotations. In the **M-step**, we train an annotator on the larger dataset aligned to our training format. This annotation process consists of three phases: *Factual Existence Judgment*, *Reference Information Extraction*, and *Hallucination Type Judgment*. As a result, we gain a **stronger annotator** with higher accuracy.

is challenging for automatic annotators due to its meticulous nature, which requires fine validation with long documents. It is also noteworthy that most self-improvement works require extra resources such as human labor or a supplementary model [36, 38]. In contrast, our pipeline is self-sufficient, relying solely on the annotator model and the initial dataset.

**Hallucination Annotation Dataset.** The development of the hallucination annotation dataset is the cornerstone for detecting hallucinations in models' output. These datasets can be used to train a hallucination detector/annotator and evaluate the hallucination level via the detector/annotator. Early works [21, 22, 26, 37, 40, 44, 48, 60, 62, 67] in this domain tended to broadly classify entire responses as either hallucinatory or not, providing a coarse-grained analysis of hallucination occurrences. Recent works [29, 47] annotate hallucinations in a more fine-grained and meticulous way. Despite this progress, these datasets, especially those having fine-grained annotations, suffer from limitations in size and scalability due to the high costs associated with the usage of human annotators or commercial models like GPT4. In addition, the difficulty of this task and the limited human annotations result in unsatisfactory performance of the automatic hallucination annotator.

**Hallucination Mitigation.** Considering the harm of hallucinations, researchers have explored various techniques for mitigating hallucinations. Techniques such as multi-task learning [23, 65], model editing [18, 31], and fine-grained RLHF [66] are proposed to suppress hallucination tendencies during training. Alternative strategies have been proposed that do not require further model training, including different decoding strategies [15, 41, 52, 55], multi-agent methods [20], and variants of the Chain-of-Thought approach involving verification or reflection [19, 32, 39, 64]. Our ANAH-v2 shows efficiency in hallucination mitigation as a re-ranker and has the potential to combine with the existing methods such as fine-grained RLHF.

## 3 Method

This paper proposes an iterative self-training framework to simultaneously scale up the hallucination dataset and improve the accuracy of the hallucination annotator. We follow the analytical hallucination annotation (§ 3.1) to annotate the hallucination sentence-by-sentence. The multi-iteration framework is theoretically grounded in the EM algorithm (§ 3.2) and involves three stages to progressively scale the dataset in multiple dimensions (§ 3.3). We also reveal how the hallucination annotators can be applied for hallucination evaluation and mitigation (§ 3.4).

## 3.1 Analytical Hallucination Annotation

The aim of a hallucination annotator is to identify hallucinations in the model responses. ANAH [29] developed a fine-grained annotation method that locates reference points in the document for each sentence and makes hallucination-type judgments, with the whole process completed in one turn of dialog. However, this hybrid task diverges from the human judgment processes and fails to clearly indicate the relationship between reference points and hallucination judgments, resulting in unsatisfactory annotation accuracy.

Instead of using the original ANAH training prompts, we developed a more reliable training method tailored to the hallucination annotation process. As depicted in the lower right part of Fig. 2, the process is outlined in three phases: (1) **Factual Existence Judgment**, where the annotator assesses whether the provided sentence contains verifiable facts. If no factual content is present, the sentence is categorized as *'No Fact'* and requires no further annotation. (2) **Reference Information Extraction**, where the annotator extracts relevant reference points from the documents related to the question and answer. (3) **Hallucination-Type Judgment**, where the annotator determines the type of hallucination based on the extracted reference points. If the sentence aligns with the references, it is classified as *'No Hallucination'*. If it contradicts the references, it is deemed a *'Contradictory Hallucination'*. If it lacks supporting evidence and cannot be verified, it is labeled as *'Unverifiable Hallucination'*. The above three phases will form a multi-turn dialogue in training data. Compared to the ANAH approach, which involves simultaneous judgments on multiple criteria, our phased process aligns more closely with human cognitive judgment processes. The detailed data format and prompts for our annotation process are in Appendix A.

## 3.2 Expectation-Maximization Algorithm

Simultaneously scaling up the dataset and improving the accuracy of the annotator can be formulated by the EM algorithm. For the input set $X$, we need to estimate two hidden variables simultaneously, the output set $Y$ and the model parameters $\theta$. Specifically, based on the task formulation in § 3.1, we define the input $x$ from the input set $X$ of the hallucination annotator consists of a question, a sentence to be annotated, and a reference document. The expected output $y$ to be estimated in the data output set $Y$ includes the factual information $f$, the key reference points $r$ from the reference document, and the type of hallucination $h$. We maximize the log-likelihood estimation of $Y$ by alternately performing the E-Step and the M-Step to update the model parameters $\theta$:

$$\theta = \arg\max_{\theta} E_{p_\theta(Y|X,\theta)} \left[\log p_\theta(X, Y \mid \theta)\right] \tag{1}$$

**E-Step.** A straightforward approach to estimating $Y$ is to use a single model to predict annotations. However, this method lacks sufficient accuracy [45]. To improve the accuracy and stability of the estimation of $Y$, we introduce the **self-consistency** method [63], which provides a more robust representation of the distribution of the $Y$. As shown in Fig. 2. For each input $x$, we perform multiple samplings to yield K independent outputs $y = \{y^1, \cdots, y^i, \cdots, y^K\}$, where the $i$-th output sample $y^i$ is composed of factual information ($f^i$), reference point ($r^i$) and hallucination type ($h^i$). We use a self-consistency metric to select the most representative sample $y^*$ among all outputs:

$$y^* = (f^*, r^*, h^*) = \text{self-consistency}(y) \tag{2}$$

During this selection process, we consider the hallucination type $h$, reference point $r$, and factual information $f$ in turn. We determine the most common hallucination type $h^*$ by tallying a **majority vote** across all samples, denoted as $h^* = \arg\max_h \sum_{i=1}^{K} \mathbb{I}(h_i = h)$. Then, we form the candidate reference set $R$ by taking the corresponding $r$ from the output containing the $h^*$. We select the most "consistent" reference point $r^*$ by comparing the cosine similarities. For each $r^i$ in $R$, we first calculate its average cosine similarity with the other elements in $R$. After that, we select the reference point $r^*$ with the highest average cosine similarity: $r^* = \arg\max_{r^i \in R}(\frac{1}{n-1}\sum_{j=1,j\neq i}^{n} \text{sim}(r^i, r^j))$. Finally, with $(r^*, h^*)$, we can uniquely select the corresponding $f^*$.

**M-Step.** Following the robust estimation in the E-step, the M-step updates the model parameters to maximize the likelihood of the selected output $y^*$. Combining Eq. 1 and Eq. 2, we formulate the parameter update strategy at iteration $t$:

$$\theta^{t+1} = \arg\max_{\theta} E_{x \sim X} \left[E_{y \sim p_{\theta^t}(y|x,\theta)} \left[\log p_\theta(x, y^* \mid \theta)\right]\right] \tag{3}$$

| Stage | # Topic | # Response | # Sentence |
|-------|---------|------------|------------|
| Stage1 | 800 | 2798 | 12188 |
| Stage2 | 800 | 46006 | 209241 |
| Stage3 | 3172 | 196930 | 822520 |

Table 1: The dataset size for ANAH-v2 in different stages, including the number of topics, model responses, and annotated sentences.

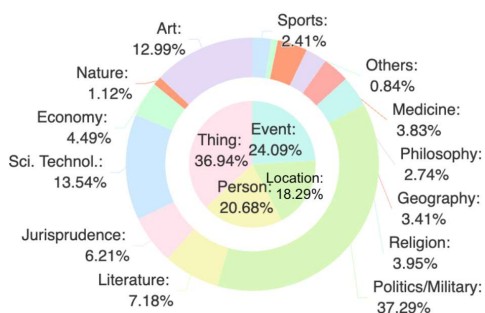

Figure 3: The topic distribution by chart of categories (inner) and domains (outer).

### 3.3 Multi-dimensional Data Scaling

Grounded in the EM algorithm, our framework operates in an iterative manner. This multi-iteration process acts as a data growth flywheel to progressively scale up the dataset in multiple dimensions, consisting of three stages:

**Stage 1: Seed Data and Basic Annotator.** We utilize ANAH dataset [29] as our seed data, which includes over 700 topics and around 4,300 LLM-generated questions and responses. For each response, ANAH provides the hallucination type for every sentence, determined through a human-in-the-loop approach. We train an initial hallucination annotator, noted as ANAH-v2 Stage1, with this seed data using the annotation method described in § 3.1.

**Stage 2: Scaling up in Response Dimension.** In Stage 1, for each question, ANAH provides responses that GPT-3.5 generates with the reference document, while InternLM-7B generates without any reference document. We first augment the dataset's model responses by collecting responses to the same existing questions from 13 additional open-source models of various sizes and series. For each model, responses were collected with and without knowledge of reference documents. The prompt details are in Appendix B. After filtering out similar model responses, these responses are annotated sentence by sentence using the self-consistency pipeline with ANAH-v2 Stage1. The newly annotated data, combined with the seed data, was used to train ANAH-v2 Stage2.

**Stage 3: Scaling up in Topic Dimension.** We expand the topic coverage along four categories: location, person, event, and thing, paralleling ANAH's configuration. For each topic, we generate several questions based on the provided reference documents (more details in Appendix B). Then, we use the same method in Stage 2 to collect responses from multiple models and annotate the response following the same procedure as in Stage 2, using ANAH-v2 Stage2 annotator. The resulting dataset, combined with data from the previous stages, is used to train the ultimate annotator version.

**Overal Statistics.** The final dataset encompasses both over ∼3k topics, ∼196k model responses, and ∼822k annotated sentences, in English and Chinese (Tab. 1). The topics cover celebrities, events, locations, and things, and span a wide array of domains, such as politics, health, and sports (Fig. 3). The statistics underscore the comprehensiveness and extensive scale of our dataset.

### 3.4 Applications

**Hallucination Evaluation.** As the accuracy of the hallucination annotators becomes satisfactory, we can apply it to automate the process of evaluating the hallucination levels of existing open-source models. After categorizing sentences into four distinct types (introduced in § 3.1), we consider type *Contradictory* and *Unverifiable Hallucination* as sentences with hallucinations, and type *No Fact* and *No Hallucination* as sentences without hallucinations. This tool enables researchers to assess the reliability and accuracy of generated texts, ensuring models can be responsibly integrated into practical applications.

**Hallucination Mitigation.** We further show a simple re-ranking strategy to mitigate the LLM's hallucinations with the annotator, whereas more advanced strategies can be explored in future research. Specifically, we adopt our annotator $\theta$ for response re-ranking. LLM first generates $N$ candidate responses $\{G_1, \cdots, G_N\}$ by top-k sampling. Then we select the best response $G^*$ with the lowest

hallucination rate over all the generated responses as below:

$$G^* = \underset{n \in \{1, \cdots, N\}}{\arg \min} \frac{|\{a_{\theta,i,n} | a_{\theta,i,n} \in \{A_C, A_U\}\}|}{L_n} \tag{4}$$

where $a_{\theta,i,n}$ is the generated annotation type by $\theta$ given the input $x_{i,n}$ including a question, the $i$-th sentence to be annotated from $G_n$, and a reference document. $A_C$ and $A_U$ means sentence type *Contradictory* and *Unverifiable Hallucination*, respectively. $L_n$ is the sentence number of $G_n$.

## 4 Experiment

### 4.1 Experimental Setup

**Implementation.** In our experimental framework, we adopt the pre-trained InternLM2-7B [7] model to fine-tune the hallucination annotator. Further implementation details can be found in Appendix C.

**Evaluation.** We use a subset of the ANAH [29] data as a test set, which is not used for training in stage 1. To assess the performance of the annotator in predicting hallucination types, we utilize **F1** and **Accuracy**. We also employ **RougeL** [42] and **BertScore** [72] to compare the generated text with gold-standard human reference in terms of gram, continuity, order and semantics.

### 4.2 Overall Results

The last 3 rows of Tab. 2 illustrate the performance of ANAH-v2 at each stage of Data Scaling in § 3.3. The performance progressively improves with the increasing dataset number (see in Tab. 1) in successive stages. This trend underscores the scalability and effectiveness of our hallucination annotation framework. Remarkably, ANAH-v2 surpasses GPT-4 with the F1 of 87.78% and the accuracy of 88.03% at Stage 2. Eventually, we achieve the F1 of 89.30% and the accuracy of 89.55% at Stage 3.

| Model | F1 ↑ | ACC ↑ | R ↑ | BERT ↑ |
|---|---|---|---|---|
| GPT-4 | 87.11 | 86.97 | 86.32 | 96.21 |
| ANAH-7B | 78.69 | 79.92 | 58.51 | 87.27 |
| ANAH-20B | 80.49 | 81.01 | 58.82 | 88.44 |
| ANAH-v2-Stage1 | 84.45 | 84.85 | 60.10 | 88.43 |
| ANAH-v2-Stage2 | 87.75 | 88.18 | 67.28 | 90.80 |
| ANAH-v2-Stage3 | 89.30 | 89.55 | 69.44 | 91.43 |

Table 2: Evaluation results for GPT4, ANAH, and ANAH-v2 at each stage, where "R" and "BERT", refer to "RougeL" and "BERTScore", respectively. [2]

Notably, the RougeL and BERTScore of GPT-4 are higher than ANAH-v2. Because GPT-4 is used for the initial pre-annotation during the construction of ANAH [29]. Subsequently, humans refine these pre-annotations, and humans tend to not change the pre-annotations. This methodology inherently aligns the final 'golden' answers closely with the outputs by GPT-4. Therefore, we tend to use "accuracy" as our primary metric because the *type judgment* is determinative of the annotation quality. For example, an annotation that wrongly judges type (low accuracy) but finds the correct *reference fragment* (high RougeL/BERTScore) remains completely unacceptable. In addition, we conduct an LLM-based evaluation to exclude the similarity due to pre-annotations in Appendix D.

We also observe that ANAH-v2 already outperforms ANAH-20B at Stage 1 (84.85% v.s. 81.01% in accuracy) with only 7B parameters, when being trained on the same hallucination corpus. This superior performance is attributed to the innovative multi-turn dialogue training strategy (§ 3.1).

### 4.3 Ablation Studies

**Impact of Self-Consistency.** To verify the effectiveness of self-consistency during inference in E-Step (introduced in § 3.2), we compare the performance of the annotator with different self-consistency settings in Tab. 3. When the annotator model with the same training data at each data scaling stage, the inference strategy with self-consistency (w/ SC) consistently outperforms without self-consistency (w/o SC), where the annotator generates only once for each input. Therefore, self-consistency improves the accuracy and stability of the estimation of hallucination annotations.

---

[2]The first three rows of data are from ANAH [29].

| Model | Train Data | Infer Strategy | F1 ↑ | ACC ↑ | R ↑ | BERT ↑ |
|---|---|---|---|---|---|---|
| ANAH-v2-Stage1 | - | w/o SC | 80.95 | 81.67 | 58.26 | 88.70 |
| | - | w/ SC | 84.45 | 84.85 | 60.10 | 88.43 |
| ANAH-v2-Stage2 | w/o SC | w/o SC | 83.80 | 83.94 | 62.93 | 89.20 |
| | w/o SC | w/ SC | 83.98 | 84.24 | 64.92 | 90.01 |
| | w/ SC | w/o SC | 84.65 | 85.15 | 61.08 | 88.47 |
| | w/ SC | w/ SC | 87.75 | 88.18 | 67.28 | 90.80 |
| ANAH-v2-Stage3 | w/o SC | w/o SC | 86.24 | 86.67 | 66.10 | 90.26 |
| | w/o SC | w/ SC | 87.78 | 88.18 | 68.18 | 91.01 |
| | w/ SC | w/o SC | 87.71 | 88.03 | 67.45 | 90.63 |
| | w/ SC | w/ SC | 89.30 | 89.55 | 69.44 | 91.43 |

Table 3: Ablation study for annotators in different self-consistency settings. Here, for *Infer Strategy*, "w/ SC" means inference with self-consistency, which is the default setting of ANAH-v2. "w/o SC" means inference without self-consistency, where the annotator generates only once for each input. For *Train Data*, "w/ SC" means the training data from the previous stage is generated by self-consistency, where the default setting of ANAH-v2, while "w/o SC" means the train data is generated without self-consistency.

| Model | Setting | F1 ↑ | ACC ↑ | R ↑ | BERT ↑ |
|---|---|---|---|---|---|
| ANAH-v2-Stage3 | progressive | 89.30 | 89.55 | 69.44 | 91.43 |
| | non-progressive | 85.88 | 86.36 | 66.10 | 90.26 |

Table 4: Ablation study for annotators trained with progressive and non-progressive data scaling. Here, "progressive" means that the training data is progressively annotated by the continually updated annotator, which is the default setting of ANAH-v2. "non-progressive" means that the training data scaling only leverages annotations generated by the basic annotator from Stage 1.

In M-Step, we train the model on data from the E-Step of the preceding iteration. We observe that when the annotator model with the same inference strategy, the model trained on self-consistently processed data (w/ SC) surpasses the performance with data generated through a single pass (w/o SC). This finding indicates that training data processed through self-consistency leads to a stronger annotator. This improvement can be attributed to the reduced distribution variance between the inferred labels and true labels.

**Impact of Progressive Data Scaling.** To assess the impact of progressive data scaling (introduced in §3.3), we compare the performance of annotators with different types of data scaling in Tab. 4. In our progressive approach, the updated annotator from Stage 2 is employed to annotate the responses from additional topics, continuously enriching the training data. Conversely, in the non-progressive approach, the basic annotator from Stage 1 is employed to generate annotations for the additional training data during Stage 3. With the same size of training data, the annotator trained on non-progressive data scaling underperforms that with our progressive data scaling, proving the effectiveness of our progressive data scaling.

**Impact of Training Strategy.** We also analyze different training strategies for annotators in different data scaling stages in Tab. 5. In our default training process, we mix the newly annotated data with old data to re-train an annotator. Alternatively, we only use the newly annotated data to further train the annotator model from the previous stage. The results demonstrate that our training strategy with mixed training data performs better than further training with new data. The integration of different data qualities across training stages improves the robustness of the annotator model.

## 4.4 Generalization Capability Analysis

We further validate the effectiveness of ANAH-v2 on other hallucination detection datasets using two third-party datasets: HaluEval [40] for English and HalluQA [14] for Chinese. Each dataset provides four components: questions, reference documents, responses, and labels indicating whether the responses contain hallucination. For each question, we let ANAH-v2 judge the type of responses containing and not containing the hallucination separately. Note that in HaluEval we only use the QA

| Model | Train Strategy | F1 ↑ | ACC ↑ | R ↑ | BERT ↑ |
|---|---|---|---|---|---|
| ANAH-v2-Stage2 | mix | 87.75 | 88.18 | 67.28 | 90.80 |
| | further | 85.50 | 85.91 | 62.15 | 89.30 |
| ANAH-v2-Stage3 | mix | 89.30 | 89.55 | 69.44 | 91.43 |
| | further | 87.73 | 86.52 | 68.58 | 91.03 |

Table 5: Ablation study for annotator in different train strategy settings. Here, "mix" means that the new data generated in the current iteration is mixed with the old data to train a new annotator, which is the default setting of ANAH-v2. "further" means that only the new data is used to further train the annotator from the previous stage.

samples and in HalluQA, we only use the samples that provide a textual reference document, which aligns with our annotator's designed setting.

The primary metric we use for evaluation is Accuracy in determining the type of response. We compare the zero-shot performance of ANAH-v2 with current SOTA results on HaluEval achieved by KnowHalu [71] and baseline results by GPT-4.

The results in Tab. 6 reveal that our annotation model achieves notable accuracies on both HaluEval and HalluQA. Remarkably, ANAH-v2-Stage3 obtains new SOTA accuracy on HaluEval (81.54%) and HalluQA (94.44%) even under a zero-shot setting, underscoring the generalization capability of ANAH-v2. Moreover, we find that ANAH-v2-Stage3 outperforms the annotators from Stage1 and Stage2, further proving the data scaling strategy effectively stabilizes performance when dealing with unfamiliar responses.

| Dataset | Model | Method | ACC ↑ |
|---|---|---|---|
| HaluEval | GPT4 | Zero-Shot | 65.05 |
| | GPT3.5 | WiKiChat [54] | 49.10 |
| | | HaluEval | 56.90 |
| | | KnowHalu | 80.30 |
| | Starling-7B | HaluEval | 61.00 |
| | | KnowHalu | 80.70 |
| | ANAH-v2-Stage1 | Zero-Shot | 79.85 |
| | ANAH-v2-Stage2 | Zero-Shot | 81.24 |
| | ANAH-v2-Stage3 | Zero-Shot | 81.54 |
| HalluQA | GPT4 | Zero-Shot | 62.81 |
| | ANAH-v2-Stage1 | Zero-Shot | 91.74 |
| | ANAH-v2-Stage2 | Zero-Shot | 92.63 |
| | ANAH-v2-Stage3 | Zero-Shot | 94.44 |

Table 6: Annotator accuracy using different models and methods on HaluEval and HalluQA. [3]

## 4.5 Application

**Hallucination Evaluation Benchmark.** Our ANAH-v2 dataset and annotator can serve as a benchmark for the hallucination levels in generated texts by existing models. As shown in Tab. 7, we evaluate the performance of various LLMs, including InternLM2 [7], Qwen1.5 [2], Baichuan2 [4], Mistral [33, 34], DeepSeek-LLM [6], and Llama2 [59], spanning different model sizes. We also offer detailed evaluation results on different languages and categories of topics to deepen our understanding.

We find that all models exhibit superior performance in English compared to Chinese, underscoring the need for further research to understand and mitigate language-dependent discrepancy. The performances of all models with reference documents are better than those without. Qwen1.5-14B achieves the lowest hallucination rate when using reference documents (5.33%) and Deepseek-67B achieves the lowest hallucination rate when reference documents are not provided (47.17%). Moreover, we find no clear trend in the performance distribution across four categories of topics. In addition, the results of different stages of annotators in Tab. A2, A3, and 7 show that there is a consistent trend and fixed biased ordering relationship between LLMs, thus confirming the reliability of our assessment method. More details are in Appendix E.

**Hallucination Mitigation.** Besides being used to measure hallucination levels, ANAH-v2 can also be used to mitigate hallucinations. We use the QA samples from HaluEval, which comprises questions and correct answers from HotPotQA [68]. We use two models InternLm2-7B and LLaMA2-7B. For each model, we generate 36 candidate responses by top-k sampling (k=40), then re-rank the responses using our annotator. To quantify the hallucination degree, we employ RougeL, BertScore, NLI, and QuestionEval. These metrics measure the congruence between the generated responses with the golden responses and/or reference documents.

---

[3]The first six rows of data are from KnowHalu [71].

| Model | Setting | Overall ↓ | Person ↓ | | Event ↓ | | Thing ↓ | | Location ↓ | |
|---|---|---|---|---|---|---|---|---|---|---|
| | | | ZH | EN | ZH | EN | ZH | EN | ZH | EN |
| InternLM2-7B | w/o Ref | 87.84 | 87.24 | 47.65 | 91.37 | 73.83 | 89.49 | 77.13 | 94.26 | 82.92 |
| | w/ Ref | 19.02 | 26.12 | 5.57 | 19.26 | 4.20 | 19.40 | 2.59 | 9.77 | 3.26 |
| InternLM2-20B | w/o Ref | 78.20 | 74.67 | 47.25 | 82.36 | 72.32 | 82.16 | 80.29 | 87.32 | 81.49 |
| | w/ Ref | 16.52 | 19.99 | 4.25 | 15.23 | 7.00 | 19.66 | 7.20 | 3.42 | 5.76 |
| Qwen1.5-7B | w/o Ref | 80.09 | 79.22 | 52.81 | 82.61 | 75.7 | 83.26 | 73.85 | 86.56 | 78.09 |
| | w/ Ref | 6.96 | 5.82 | 2.77 | 5.27 | 3.76 | 9.70 | 3.69 | 4.90 | 4.40 |
| Qwen1.5-14B | w/o Ref | 68.82 | 65.63 | 44.91 | 70.25 | 68.24 | 72.36 | 70.76 | 73.37 | 72.69 |
| | w/ Ref | 5.33 | 5.01 | 1.23 | 4.56 | 2.02 | 7.38 | 2.70 | 2.53 | 2.00 |
| Qwen1.5-72B | w/o Ref | 61.62 | 56.49 | 29.76 | 61.62 | 56.78 | 67.42 | 62.92 | 67.97 | 64.36 |
| | w/ Ref | 15.89 | 19.27 | 4.62 | 13.85 | 3.18 | 18.99 | 3.80 | 5.50 | 4.26 |
| Baichuan2-7B | w/o Ref | 73.99 | 72.13 | 44.99 | 75.77 | 65.98 | 76.84 | 73.01 | 71.51 | 74.17 |
| | w/ Ref | 43.68 | 61.71 | 64.56 | 37.87 | 26.41 | 35.4 | 29.17 | 54.3 | 14.51 |
| Baichuan2-13B | w/o Ref | 69.85 | 67.02 | 41.24 | 71.63 | 63.13 | 73.32 | 66.2 | 68.77 | 71.35 |
| | w/ Ref | 38.39 | 58.86 | 60.53 | 43.20 | 21.9 | 25.74 | 17.81 | 28.99 | 7.23 |
| Mistral-7B | w/o Ref | 85.40 | 89.98 | 52.32 | 87.03 | 72.33 | 87.41 | 71.97 | 91.19 | 77.25 |
| | w/ Ref | 30.24 | 42.66 | 22.83 | 30.85 | 13.77 | 26.02 | 27.15 | 42.11 | 7.23 |
| Mistral-8x7B | w/o Ref | 76.12 | 80.96 | 30.75 | 76.78 | 55.32 | 83.32 | 61.61 | 87.28 | 65.51 |
| | w/ Ref | 7.95 | 8.29 | 2.78 | 6.17 | 5.84 | 9.91 | 7.94 | 3.99 | 6.63 |
| Deepseek-7B | w/o Ref | 64.46 | 65.98 | 39.62 | 67.59 | 60.69 | 69.51 | 56.15 | 69.29 | 59.15 |
| | w/ Ref | 23.02 | 6.73 | 44.95 | 28.38 | 4.92 | 25.00 | 24.25 | 18.18 | 12.61 |
| Deepseek-67B | w/o Ref | 47.17 | 54.91 | 15.81 | 46.28 | 31.48 | 65.57 | 34.23 | 59.96 | 36.02 |
| | w/ Ref | 12.05 | 12.61 | 4.17 | 9.52 | 2.00 | 15.79 | 13.36 | 18.65 | 8.33 |
| Llama2-7B | w/o Ref | 84.22 | 88.36 | 52.00 | 84.95 | 74.18 | 92.48 | 77.89 | 89.84 | 78.91 |
| | w/ Ref | 58.16 | 82.5 | 10.64 | 76.96 | 10.00 | 64.72 | 12.33 | 69.75 | 20.48 |
| Llama2-13B | w/o Ref | 78.84 | 80.26 | 43.18 | 81.88 | 70.25 | 87.85 | 70.52 | 84.44 | 73.94 |
| | w/ Ref | 52.17 | 79.43 | 14.81 | 47.85 | 4.00 | 49.59 | 11.72 | 77.50 | 27.53 |

Table 7: Hallucination rate of open-source models according to ANAH-v2 annotator and dataset.

| Model | Setting | QuestEval ↑ | NLI ↑ | BERT ↑ | RourgeL ↑ |
|---|---|---|---|---|---|
| LLaMA2-7B | baseline | 37.84 | 31.25 | 83.76 | 19.34 |
| | re-rank | 38.50 | 36.03 | 84.45 | 21.92 |
| InternLM2-7B | baseline | 37.33 | 25.00 | 83.57 | 20.55 |
| | re-rank | 38.89 | 37.01 | 84.57 | 22.39 |

Table 8: Evaluation results for hallucination mitigation with LLaMA2-7B and InternLM2-7B on HaluEval. Here, "baseline" means the direct generation results, and "re-rank" means the results with our re-ranking mitigation method.

Results in Tab. 8 show a clear reduction of hallucination levels after the re-ranking process via our annotator. For instance, the NLI metric for LLaMA2-7B shows a notable increase, rising from 25.00% to 37.01%. This suggests that the application of our annotative approach can significantly mitigate the issue of hallucinations in language model outputs.

## 5   Conclusion and Future Work

In this paper, we aim to explore a scalable framework for the oversight of LLM hallucinations. Through iterative self-training, we progressively expand the diversity and scale of the dataset and improve the accuracy of the hallucination annotator. The finally obtained ANAH-v2, for the first time, outperforms GPT-4 in various hallucination detection benchmarks with only 7B parameters and obtains superior zero-shot performance on third-party hallucination detection benchmarks. ANAH-v2 not only provides an automatic hallucination evaluation benchmark with the scaled dataset, which paves the way for future research on hallucination mitigation but also exhibits potential in

hallucination mitigation by the simple re-ranking strategy. We believe ANAH-v2 can also benefit more hallucination mitigation strategies such as fine-grained RLHF.

With the large-scale dataset as seed data, future work can explore creating hallucination annotation data in other NLG tasks such as dialogue generation. Another direction is to improve the generalizability of the annotator across different languages, tasks, and topics.

## Acknowledgement

We thank the anonymous reviewers and area chair for their helpful comments. This project is funded in part by the Hong Kong Generative AI Research and Development Center (HKGAI) under the Innovation and Technology Commission (ITC)'s InnoHK. Dahua Lin is a PI of HKGAI under the InnoHK. This project is also supported by the Shanghai Artificial Intelligence Laboratory. The authors would like to thank Zehui Chen and Kuikun Liu for their valuable suggestions and comments.

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

## A  Training Prompt

As described in § 3.1, our annotation process consists of three phases: (1) Factual Existence Judgment via the prompt in Fig. A1, (2) Reference Information Extraction via the prompt in Fig. A2 (3) Hallucination-Type Judgment via the prompt in Fig. A3.

## B  Data Scaling Details

As described in § 3.3, we collect model responses via Fig. A5. The open-source models include InternLM2(7B&20B) [7], Baichuan2 (7B&13B) [4], LLama2 (7B&13B) [59], Qwen1.5 (7B&14B&72B) [2], Deepseek (7B&67B) [6], and Mistral (7B&7×8B) [33, 34].

We automate the topic selection based on occurrence frequency via Google Ngram Viewer [4] and retrieve corresponding reference documents from pre-training databases [27].

We generate questions on each topic via Fig. A4.

## C  Implementation Details

In our experimental framework, we adopt the pre-trained InternLM2-7B [7] model to fine tuning the hallucination annotator.

In **E-Step**, we generate responses by implementing sampling via the LMDeploy library [17]. During each iteration, we generate 32 candidate responses per input and apply a self-consistency quality control mechanism to them. The decoding strategy involves the top-k (k = 40) sampling with a temperature of 0.8.

In **M-Step**, we train the annotator model with the following settings and hyper-parameters: the epoch is 1, the learning rate is 1e-5, and the AdamW optimizer is with a linear scheduler, the maximum sequence length is set to 32k. Additionally, following the configuration in ANAH [29], we perform a multi-task setting where additional tasks such as dialogue generation from ShareGPT [49] and Dolly [16] are integrated with the fine-grained hallucination annotation. Our model is trained on 32 NVIDIA A100 GPUs.

## D  Addtional Evaluation for the Overall Results

To exclude the similarity due to the impact of pre-annotations described in § 4.2, we conduct an LLM-based evaluation to assess the consistency of generated reference points with the source document. Specifically, we followed the prompt in FactScore [46], which aims to clarify whether the generated reference points are supported by the given source document, rather than simply calculating the similarity between them using metrics such as RougeL or BERTScore. We employ the InternLM2-7B-Chat [7] as the estimator.

| Model | Score |
|---|---|
| GPT-4 | 84.39 |
| ANAH-7B | 80.60 |
| ANAH-20B | 81.51 |
| ANAH-V2-Stage1 | 83.63 |
| ANAH-V2-Stage2 | 84.54 |
| ANAH-V2-Stage3 | 86.36 |

Table A1: The score assessing the consistency of generated reference points with the source documents.

As shown in Tab. A1, the results indicate that the reliability of our model's generated reference points progressively improves and ultimately exceeds that of GPT4. This trend is consistent with F1 and ACC in Tab 2.

## E  Hallucination Evaluation

To assess the reliability of our hallucination annotator, we measure the hallucination levels of the above LLMs using the annotator from different stages. Tab. A2, A3, and 7 show the results measured by annotator ANAH-v2 from Stage 1, 2, and 3, respectively. The trends in these three tables are

---

[4]https://books.google.com/ngrams/

consistent where Qwen1.5-14B achieves the lowest hallucination rate with reference documents and DeepseekLM-67B achieves the lowest hallucination rate without reference documents. This consistency and fixed biased ordering relationship between LLMs confirm the reliability of our assessment method.

# F  Limitation

Although this study presents a novel multi-iteration self-training framework for the scalable oversight of LLM hallucinations and achieves significant improvements in hallucination annotation, there are some limitations.

Despite the progressive scaling and increasing accuracy of the hallucination annotator, there may still exist a non-negligible margin of error in the annotations. This margin could affect the convergence of the model and the quality of the final hallucination annotator. Furthermore, the success of our framework is measured largely by its performance on our own dataset and other benchmarks such as HalluEval and HalluQA. However, these datasets might not encompass the full spectrum of real-world scenarios where hallucinations pose a problem. Lastly, this work primarily uses InternLM2-7B as the backbone of the hallucination annotator. Other different underlying models and different numbers of parameters are not explored.

In addition, the EM algorithm, which is the theoretical foundation of our framework, may also introduce some problems. For example, the EM algorithm is sensitive to initial conditions, which would impact the convergence process. Although we employ many methods to ensure the stability of the training process, such as selecting a high-quality, human-labeled hallucination dataset as the seed and using a progressive scaling strategy, we cannot claim that we have eventually converged to a globally optimal solution. Moreover, the iterative EM algorithm requires computational effort. Our method uses 32 A100 GPUs to iteratively train the 7B model. It took approximately 100 hours for inference and training. Based on the price of the computing platform Lambda (1.29 USD per GPU per hour), it costs 4,128 USD. However, using the "manual + GPT4-assisted" annotation model, as described in ANAH [29] (0.9 USD and 20 minutes per annotation), it would take 177,237 USD and 65,643 hours to reach the size of the dataset in our work. So we believe our method is a better trade-off between computing resources and labour+API costs, which is acceptable.

# G  Broader Impacts

By exploring the hallucination annotation and mitigation in LLMs, this paper contributes to the development of more reliable and trustworthy AI technologies. Our innovative multi-iterative self-training framework significantly reduces the reliance on expensive and time-consuming manual annotations by automating the hallucination detection process. Our hallucination annotator offers a benchmark for the research community evaluating the hallucination levels of existing open-source models. Additionally, we provide a large-scale and diverse dataset from which the broader research community can benefit, fostering further innovation and study in this domain.

**English Prompt:**
You will act as a fact checker, and I will provide you with a question and a corresponding partial answer. Your task is to determine whether the content of the answer contains verifiable facts.
## Judgment Criteria:
- Verifiable Facts: Specific, objective points of information that can be verified through data, research results, or other reliable sources. Examples include statistical data, historical events, scientific laws, and specific case studies.
- Non-factual Descriptions: Personal opinions, subjective judgments, or unverifiable statements.
## Task Process:
1. Carefully read the question, which is as follows: {*question*}
2. Carefully read the partial answer, which is as follows: {*annotation*}
3. Conduct the Analysis: Based on the above judgment criteria, determine if the answer contains verifiable facts.
- If there are no verifiable facts in the answer, output *"<No Facts>"*.
- If there are verifiable facts in the answer, output *"<Facts Present>"*.

**Chinese Prompt:**
你将作为一个事实判断器，我会给你提供一个问题和一个针对该问题的部分回答，你的任务是判断回答中的内容是否存在可以判断的事实。
## 判断标准：
- 可以判断的事实：具体的、客观的信息点，这些信息可以通过数据、研究结果或其他可靠来源进行验证。例如，统计数据、历史事件、科学定律、具体案例等。 - 非事实描述：个人意见、主观判断或无法验证的声明。
## 任务流程：
1. 仔细阅读问题，问题如下： {*question*}
2. 仔细阅读回答，部分回答如下： {*annotation*}
3. 进行分析：根据上述判断标准，判断回答中是否包含可以判断的事实。
- 如果回答中不存在可以判断的事实，则输出 "<无事实>"。 - 如果回答中存在可以判断的事实，则输出 "<有事实>"。

Figure A1: Prompts for factual existence judgment.

**English Prompt:**
You will act as an information extractor. I will provide you with a question, a related reference document, and a partial answer to that question. Your task is to extract information from the reference document that is relevant to the question and answer.
## Operational Steps:
1. Carefully read the question, which is as follows: {*question*}
2. Carefully read the partial answer, which is as follows: {*annotation*}
3. Analyze the Reference Document: Identify information most relevant to the question and answer. This information may be completely the same, partially similar, or conflicting with the content of the answer. The reference document is as follows: {*reference*}
4. List the Relevant Information: List all the relevant information found in order, separated by <SEP> if there are multiple pieces of information.
5. Output When No Information Is Found: If no relevant information is found, output *<No Reference Information>*.

**Chinese Prompt:**
你将作为一个信息提取器，我将给你提供一个问题、一份相关的参考文档，以及一个针对该问题的部分回答，你的任务是从参考文档中提炼出与问题和回答相关的信息。
## 操作步骤:
1. 仔细阅读问题，问题如下：{*question*}
2. 仔细阅读回答，部分回答如下：{*annotation*}
3. 分析参考文档：找出与问题和回答最相关的信息，这些信息可能与回答内容完全相同、部分相同，或存在冲突。参考文档如下：{*reference*}
4. 列出相关信息：按顺序列出所有发现的相关信息，如果有多条信息的话以 <SEP> 作为分隔。
5. 无相关信息时输出：如果没有找到相关信息，请输出<无参考信息>。

Figure A2: Prompts for reference information extraction.

**English Prompt:**
You will act as a 'Hallucination' annotator. I will provide you with a question, a partial answer to that question, and related reference points. You need to determine whether the provided answer contains any hallucinatory content and annotate the type of hallucination.

'Hallucination' refers to content that contradicts the reference points or is unsupported by them.

## Judgment Criteria:

1. No Hallucination: If the answer is completely consistent with the reference points and does not introduce any contradictory information, output: *<No Hallucination>*.

2. Contradiction: If the answer clearly contradicts the reference points, output: *<Contradictory>*.

3. Unverifiable: If the answer contains information not mentioned in the reference points and cannot be supported or verified by them, output: *<Unverifiable>*.

## Task Process:

1. Carefully read the question, which is as follows: {*question*}

2. Carefully read the partial answer, which is as follows: {*annotation*}

3. Carefully read the reference points, which are as follows: {*reference*}

4. Conduct the analysis: Based on the above judgment criteria, determine if the answer contains hallucinations and output the type of hallucination.

**Chinese Prompt:**
你将作为一个'幻觉'标注器，我将会给你提供一个一个问题，一个针对该问题的部分回答和相关的参考要点。你需要判断提供的回答中是否含有幻觉性内容，并标注幻觉类型。

'幻觉'指的是与参考要点相矛盾或在参考要点中没有依据的内容。

## 判断准则：

1. 无幻觉：如果回答与参考要点完全一致，且没有引入与参考要点相矛盾的信息，请输出：<无幻觉>。

2. 矛盾：如果回答内容与参考要点存在明显矛盾，请输出：<矛盾>。

3. 无法验证：如果回答包含的信息在参考要点中没有提及，且无法从参考要点中得到支持或验证，请输出：<无法验证>。

## 任务流程：

1. 仔细阅读问题，问题如下：{*question*}

2. 仔细阅读回答，部分回答如下：{*annotation*}

3. 仔细阅读参考要点，参考要点如下：{*reference*}

4. 进行分析：根据上述判断标准，判断回答中是否包含幻觉，并输出幻觉类型。

Figure A3: Prompts for hallucination type judgment.

**English Prompt:**
I would like you to act as a question generator. I will provide references and you will generate 10 questions about "{topic}" based on the reference. The specific requirements are as follows:
1. the questions can be fully answered based only on the reference document, i.e. the answers to the questions are fully contained in the reference document. The questions should be objective and not too subjective or open-ended.
2. the 10 questions should be of as many different types as possible, e.g. what, when, where, why. Questions can be asked from different perspectives, e.g. descriptions, explanations, reasons, etc. Ensure that the questions are of different types and cover all aspects of the information.
3. 10 questions can cover different levels of knowledge, from general, basic knowledge to more specialized, complex subject knowledge or domain knowledge.
4. have only one question per item.
Reference: {reference document}
Please list the 10 questions directly based on the above reference without any explanation:

**Chinese Prompt:**
我希望你充当一个问题生成器。我将提供参考资料，你将根据资料生成关于"{topic}"的10个问题。具体要求如下：
1. 只根据参考资料，完全可以回答问题，即问题的答案完全包含在参考资料中。问题要客观，不要太过主观和开放。
2. 10个问题尽量是不同类型的，比如：什么、何时、何地、为什么。问题可以从不同的角度出发，例如描述、解释、原因等。确保问题类型多样，覆盖资料的各个方面。
3. 10个问题可以涉及不同层次的知识，从常识性、基本性的知识，到更专业化、复杂化的学科知识或领域知识。
4. 每条只有一个问题。
参考资料：{reference document}
请根据以上参考资料，不做说明直接列出10个问题：

Figure A4: Prompts for question generation.

**English Prompt:**
Reference document: {reference document}
Please answer the question based on the above reference: {question}

**Chinese Prompt:**
参考资料：{reference document}
请根据以上参考资料，回答问题：{question}

Figure A5: Prompts for answering.

| Model | Setting | Overall ↓ | Person ↓ | | Event ↓ | | Thing ↓ | | Location ↓ | |
|-------|---------|-----------|----------|----------|---------|---------|---------|---------|---------|---------|
| | | | ZH | EN | ZH | EN | ZH | EN | ZH | EN |
| InternLM2-7B | w/o Ref | 74.8 | 94.03 | 40.74 | 86.42 | 67.68 | 93.83 | 75.03 | 86.24 | 53.03 |
| | w/ Ref | 13.09 | 18.94 | 5.48 | 33.93 | 8.77 | 16.03 | 2.7 | 18.09 | 4.85 |
| InternLM2-20B | w/o Ref | 63.94 | 91.87 | 38.01 | 75.24 | 71.76 | 90.04 | 78.02 | 78 | 54.86 |
| | w/ Ref | 12.84 | 29.76 | 5.65 | 29.24 | 12.72 | 13.78 | 7.67 | 16.09 | 12.62 |
| Qwen1.5-7B | w/o Ref | 64.04 | 87.68 | 42.61 | 70.86 | 65.62 | 88.05 | 72.62 | 79.91 | 54.22 |
| | w/ Ref | 6.92 | 8.62 | 4.3 | 16.32 | 10.57 | 8.86 | 3.44 | 9.46 | 6.54 |
| Qwen1.5-14B | w/o Ref | 55.88 | 71.83 | 36.81 | 59.78 | 67.15 | 79.94 | 69.02 | 67.03 | 51.62 |
| | w/ Ref | 5.96 | 10 | 2.29 | 12.88 | 2.93 | 13.19 | 2.02 | 7.77 | 6.67 |
| Qwen1.5-72B | w/o Ref | 49.25 | 67.67 | 25.47 | 56.33 | 58.88 | 77.72 | 60.79 | 62.81 | 41.98 |
| | w/ Ref | 12.72 | 11.35 | 7.13 | 27.87 | 10.91 | 15.63 | 4.39 | 17.78 | 9.77 |
| Baichuan2-7B | w/o Ref | 63.19 | 77.99 | 39.49 | 70.56 | 61.66 | 79.26 | 71.72 | 72.7 | 52.71 |
| | w/ Ref | 52.38 | 18.02 | 64.27 | 60.78 | 37.38 | 29.17 | 27.71 | 43.61 | 23.55 |
| Baichuan2-13B | w/o Ref | 57.66 | 70.66 | 32.95 | 66.52 | 63.04 | 79.17 | 65.14 | 70.45 | 47.62 |
| | w/ Ref | 46.47 | 12.2 | 52.02 | 63.41 | 33.15 | 40.99 | 16.58 | 44.78 | 42.96 |
| Mistral-7B | w/o Ref | 70.86 | 92.31 | 43 | 89.63 | 67.09 | 87.6 | 71.25 | 87.11 | 47.99 |
| | w/ Ref | 32.22 | 10.77 | 23.45 | 48.37 | 17.74 | 43.8 | 27.8 | 27 | 30.29 |
| Mistral-8x7B | w/o Ref | 55.72 | 82.39 | 26.16 | 77.09 | 54.9 | 90.51 | 60.17 | 80.96 | 42.86 |
| | w/ Ref | 8.17 | 9.45 | 3.7 | 14.92 | 7.06 | 14.57 | 7.69 | 7.67 | 6.41 |
| Deepseek-7B | w/o Ref | 50 | 62.26 | 32.38 | 63.85 | 62.81 | 77.99 | 54.9 | 68.51 | 50.56 |
| | w/ Ref | 23.1 | 11.54 | 43.94 | 22.97 | 8.2 | 26.19 | 24.25 | 17.8 | 13.51 |
| Deepseek-67B | w/o Ref | 33.68 | 52.86 | 17.89 | 57.79 | 37.41 | 72.91 | 33.93 | 64.62 | 33.33 |
| | w/ Ref | 13.4 | 9.91 | 10 | 12.7 | 2 | 11.84 | 10.53 | 21.03 | 15 |
| Llama2-7B | w/o Ref | 67.81 | 90.22 | 42.44 | 88.27 | 70.81 | 94.44 | 76.28 | 91.84 | 56.95 |
| | w/ Ref | 50.65 | 66.67 | 13.83 | 73.04 | 11.76 | 54.3 | 13.7 | 60.5 | 21.43 |
| Llama2-13B | w/o Ref | 62.69 | 84.73 | 36.43 | 85.38 | 67.01 | 87.09 | 69.6 | 90.07 | 51.3 |
| | w/ Ref | 46.59 | 62.86 | 14.81 | 50.54 | 4.37 | 13.1 | 44.66 | 73.75 | 28.65 |

Table A2: Hallucination rate of open-source models according to ANAH-v2-Stage1.

| Model | Setting | Overall ↓ | Person ↓ | | Event ↓ | | Thing ↓ | | Location ↓ | |
|---|---|---|---|---|---|---|---|---|---|---|
| | | | ZH | EN | ZH | EN | ZH | EN | ZH | EN |
| InternLM2-7B | w/o Ref | 75.83 | 95.28 | 42.49 | 84.58 | 70.88 | 94.79 | 77.13 | 86.2 | 57.73 |
| | w/ Ref | 12.14 | 18.18 | 4.74 | 28.59 | 9.86 | 15.69 | 2.59 | 17.47 | 5.22 |
| InternLM2-20B | w/o Ref | 65.8 | 92.82 | 39.75 | 76.74 | 75.19 | 90.95 | 80.29 | 79.71 | 57.24 |
| | w/ Ref | 12.13 | 28.57 | 5.82 | 25.2 | 13.15 | 12.64 | 7.2 | 15.4 | 13.46 |
| Qwen1.5-7B | w/o Ref | 65.38 | 86.23 | 44.07 | 68.66 | 71.35 | 89.71 | 73.85 | 80.68 | 56.72 |
| | w/ Ref | 6.28 | 6.03 | 3.81 | 10.39 | 11.38 | 9.11 | 3.69 | 7.8 | 7.07 |
| Qwen1.5-14B | w/o Ref | 57.42 | 71.83 | 38.14 | 56.28 | 71.52 | 81.31 | 70.76 | 69.81 | 54.07 |
| | w/ Ref | 4.92 | 5 | 1.06 | 9.83 | 2.51 | 12.47 | 2.7 | 7.01 | 6.44 |
| Qwen1.5-72B | w/o Ref | 50.17 | 66.17 | 25.92 | 52.74 | 60.24 | 79.45 | 62.92 | 63.9 | 46.34 |
| | w/ Ref | 11.46 | 10.64 | 6.09 | 25.09 | 11.42 | 15.81 | 3.8 | 15.87 | 8.94 |
| Baichuan2-7B | w/o Ref | 63.47 | 78.62 | 39.22 | 67.77 | 64.13 | 80.74 | 73.01 | 73.65 | 54.89 |
| | w/ Ref | 53.25 | 17.12 | 65.63 | 60.61 | 39.08 | 30.87 | 29.17 | 43.25 | 25.09 |
| Baichuan2-13B | w/o Ref | 58.4 | 73.05 | 33.42 | 64.06 | 67.5 | 80.85 | 66.2 | 70.45 | 50.83 |
| | w/ Ref | 47.48 | 12.2 | 54.36 | 61.66 | 35.91 | 42.46 | 17.81 | 44.78 | 42.26 |
| Mistral-7B | w/o Ref | 71.47 | 93.01 | 43.53 | 89.82 | 68.15 | 88.06 | 71.97 | 87.75 | 49.6 |
| | w/ Ref | 32.04 | 11.28 | 23.51 | 46.9 | 18.28 | 44.49 | 27.15 | 26.91 | 32.15 |
| Mistral-8x7B | w/o Ref | 56.91 | 84.09 | 27.4 | 76.25 | 57.26 | 91.35 | 61.61 | 82.25 | 45.11 |
| | w/ Ref | 7.86 | 8.66 | 2.69 | 13.39 | 7.66 | 14.3 | 7.94 | 8.12 | 8.05 |
| Deepseek-7B | w/o Ref | 51.09 | 64.15 | 33.43 | 61.41 | 65.51 | 79.43 | 56.15 | 69.55 | 53.6 |
| | w/ Ref | 23.35 | 7.69 | 45.96 | 22.97 | 4.92 | 25.79 | 25.5 | 17.8 | 13.51 |
| Deepseek-67B | w/o Ref | 33.4 | 60 | 17 | 53.44 | 38.15 | 73.47 | 34.23 | 63.82 | 34.7 |
| | w/ Ref | 11.53 | 9.01 | 5 | 9.52 | 2 | 14.47 | 11.74 | 19.44 | 8.33 |
| Llama2-7B | w/o Ref | 69.26 | 91.3 | 44.15 | 89.3 | 74.65 | 94.99 | 77.89 | 93 | 58.01 |
| | w/ Ref | 55.47 | 78.33 | 11.7 | 74.02 | 10.34 | 61.43 | 11.64 | 67.23 | 19.05 |
| Llama2-13B | w/o Ref | 63.59 | 83.97 | 37.17 | 85.48 | 68.69 | 87.4 | 70.52 | 91 | 53.5 |
| | w/ Ref | 50.7 | 75.43 | 12.35 | 4.18 | 13.1 | 50 | 47.4 | 78.75 | 27.53 |

Table A3: Hallucination rate of open-source models according to ANAH-v2-Stage2.

