# OpenReview forum: "ANAH-v2: Scaling Analytical Hallucination Annotation of Large Language Models"
_NeurIPS.cc/2024/Conference — NeurIPS 2024 poster_

### Official Review · Reviewer_i3ot · 2024-06-29

**Soundness:** 3
**Presentation:** 4
**Contribution:** 3
**Rating:** 6
**Confidence:** 4

**Summary:**

The paper proposes to augment hallucination annotation dataset and improve the performance of the hallucination annotator simultaneously in an iterative self-training framework. During each iteration, they use a Expectation-Maximization Algorithm for data annotation and hallucination annotator training. The trained hallucination annotator can be further used for downstream tasks such hallucination evaluation and hallucination mitigation. Extensive experiments were performed to show the superiority in hallucination detection and the effectiveness of the proposed approach.

**Strengths:**

- The paper is very easy and pleasant to read. Each section and their subsections are well connected and explained without adding much redundant content. Figure 2 is especially illustrative.
- I like how they explain a rather practical work in a theoretical manner using all the math equations in Section 3. It makes the work more sound with less confusions.
- Although the model name is ANAH-v2, it actually brings big novelty compared to the original ANAH model. The model is in a self-training manner and the training prompt is expanded to do three tasks.
- Large amount of experiments show impressive performance on hallucination detection. The thorough ablation study also shows the importance of each component.

**Weaknesses:**

- My biggest concern for this paper is that the model essentially does three things: Factual Existence Judgment, Reference Information Extraction, and Hallucination-Type Judgment. Each step relies on the results from previous step. For example, if the extracted reference information is limited, it will greatly impact the hallucination judgement as well. The experiments mainly focus on the third step without discussions on the previous two steps. So I’m not certain how stable the model actually is.
- The hallucination mitigation is a little weak, it basically generates multiple responses and select the response with the least portion of hallucination sentences. There are multiple other ways to mitigate hallucinations that are not compared in Table 8.

**Questions:**

- Typo in ine 76-77, "we reduce the hallucination of the final LLM generations from 25% to 37%."
- Line 156, when you use the majority vote to select the the most common hallucination type, wouldn't it be dominanted by "No Hallucination"? As you are also showing in Table 7, the hallucination rate for InternLM2 is less than 20%. If you generate n candidates (depending on how big your n is) and use a majority votes, not many hallucination types will be selected.
- How many iterations did you do to get the dataset in Table 1?
- In Table 2, any discussions on GPT4 having much better RougeL and BertScore?

**Limitations:**

The limitations are discussed in Appendix which actually addresses some of the questions I was gonna ask, such as more benchmarks or more backbone models for the hallucination annotator.

---

> ### Author Rebuttal · Authors · 2024-08-07
>
> Thank you for your constructive comment. Following are our responses to each individual comment (which are highlighted in italics).
>
> ### **Response to Weakness1 about stability of the annotation steps:**
>
> > *My biggest concern for this paper is that the model essentially does three things: Factual Existence Judgment, Reference Information Extraction, and Hallucination-Type Judgment. Each step relies on the results from previous step. For example, if the extracted reference information is limited, it will greatly impact the hallucination judgement as well.*
>
> We acknowledge that there exists the possibility that such previous steps may have influenced the final judgement. Therefore, in addition to majority voting on the final step of data construction, we also perform consistency checks (lines 157-161) on the previous steps to ensure quality.
>
> Meanwhile, we utilize RougeL, Bertscore to evaluate the performance of Factual Existence Judgment (Step1) and Reference Information Extraction (Step2), and we find that these two metrics in Table 2 increase steadily as the iteration progresses. These evaluations can show the stability and effectiveness of previous steps.
>
> In addition, Hallucination-Type Judgment (Step 3) relies on the results from the previous steps. Therefore, the results for Step 3 (F1 and ACC in Table 2) can also reflect the robustness and accuracy of the previous steps.
>
> ### **Response to Weakness2 about mitigation:**
>
> > *The hallucination mitigation is a little weak, it basically generates multiple responses and select the response with the least portion of hallucination sentences. There are multiple other ways to mitigate hallucinations that are not compared in Table 8.*
>
> Our primary focus is on automatically scaling up the annotation dataset and building annotators.
> The mitigation section aims to show the potential of our annotator for mitigation, rather than to achieve SOTA results.
>
> Therefore, we only used a simple re-ranking strategy to mitigate the hallucination of LLM, and the promising results in Table 8 proved that our annotator could be used for hallucination mitigation.
>
> In the future, we will explore methods to apply our annotators in mitigation, such as RLAIF.
>
> ### **Response to Question1 about typo error:**
>
> > *Typo in ine 76-77, "we reduce the hallucination of the final LLM generations from 25% to 37%."*
>
> We use "reduce" because the metric NLI described here is inversely related to the level of hallucinations. For clarification, we will change the presentation to "we reduce hallucination, with the NLI metric increasing from 25% to 37% on HaluEval" in the next version.
>
> ### **Response to Question2 about balance:**
>
> > *Line 156, when you use the majority vote to select the the most common hallucination type, wouldn't it be dominanted by "No Hallucination"? As you are also showing in Table 7, the hallucination rate for InternLM2 is less than 20%. If you generate n candidates and use a majority votes, not many hallucination types will be selected.*
>
> To clarify, the hallucination rate of InternLM2 less than 20% in Table 7 is achieved at settings that provide reference when generating response.
>
> In the setting of "w/o reference", the hallucination rate of InternLM2 is ~80% and the responses both with and without reference are mixed together to construct the dataset. Therefore, although the majority voting may enhance such bias in the single setting, the overall dataset is balenced. For example, in the training data at Stage 2, the ratio of hallucinations to non-hallucinations is **52.53 : 47.47**.
>
> In addition, the results in Table 3 show our majority-vote method improves accuracy.
>
> We will add a corresponding discussion in the next version.
>
> ### **Response to Question3 about iteration:**
>
> > *How many iterations did you do to get the dataset in Table 1?*
>
> 3 iterations
>
> ### **Response to Question4 about GPT4 performance:**
>
> > *In Table 2, any discussions on GPT4 having much better RougeL and BertScore?*
>
> As we mentioned in footnote 2, GPT-4 pre-annotation in ANAH makes the RougeL and BERTScore higher than the zero-shot setting in this work. In order to analyze the causes of this phenomenon more clearly.
>
> During the construction of ANAH [1],  GPT-4 is used for the initial pre-annotation. Subsequently, humans refine these pre-annotations, and humans tend to not change the pre-annotations. This methodology inherently aligns the final 'golden' answers closely with the outputs by GPT-4.
>
> In addition, we added an LLM-based Evaluation to exclude the similarity due to pre-annotations. Specifically, we use FactScore [4] to assess the consistency of generated reference points with the source document.
>
> Below is Table 2 from our paper, which contains the newly added metric FactScore (column 3). The results of the FactScore indicates that the reliability of our model's generated reference points progressively improves and ultimately exceeds that of GPT4. This trend is consistent with F1 and ACC, reflecting the reliability of the FactScore.
>
> | Model          | F1    | ACC   | **FactScore** | RougeL | BertScore |
> |----------------|-------|-------|---------------|--------|-----------|
> | GPT-4          | 87.11 | 86.97 | **84.39**     | 86.32  | 96.21     |
> | ANAH-7B        | 78.69 | 79.92 | **80.60**     | 58.51  | 87.27     |
> | ANAH-20B       | 80.49 | 81.01 | **81.51**     | 58.82  | 88.44     |
> | ANAH-V2-Stage1 | 84.45 | 84.85 | **83.63**     | 60.10  | 88.43     |
> | ANAH-V2-Stage2 | 87.75 | 88.18 | **84.54**     | 67.28  | 90.80     |
> | ANAH-V2-Stage3 | 89.30 | 89.55 | **86.36**     | 69.44  | 91.43     |
>
> We will add a corresponding discussion and evaluation results in the next version.
>
> [1] Ji Z, Gu Y, et al. ANAH: Analytical Annotation of Hallucinations in Large Language Models[J]. ACL, 2024.
>
> [2] Min S, Krishna K, Lyu X, et al. Factscore: Fine-grained atomic evaluation of factual precision in long form text generation[J]. arXiv preprint arXiv:2305.14251, 2023.

---

> ### Author Response · Authors · 2024-08-14
> **Please let us know if your concerns have been addressed**
>
> Dear Reviewer i3ot,
>
> We would like to thank you for the thoughtful and constructive feedback and appreciate that you agree on the strengths of our paper. During the rebuttal, **we have provided more details and analysis to address your concerns.** As the discussion phase is nearing its end, we are warmly concerned whether our rebuttal addresses your concerns.
>
> **It would be appreciated if you could raise your score on our paper if we address your concerns.** We thank you again for your effort in reviewing our paper.
>
> Best regards,
>
> Authors of Submission 10034.

---

### Official Review · Reviewer_5wgD · 2024-07-11

**Soundness:** 2
**Presentation:** 3
**Contribution:** 3
**Rating:** 5
**Confidence:** 4

**Summary:**

This paper introduces an iterative self-training framework to address hallucinations in large language models (LLMs), enhancing the accuracy of annotators and scaling up hallucination detection datasets. The framework utilizes the Expectation Maximization algorithm to progressively improve the hallucination annotator's performance by annotating a scaled dataset and training a more accurate annotator in each iteration. Experimental results demonstrate that the enhanced annotator surpasses GPT-4 in hallucination detection, achieving state-of-the-art results on HaluEval and HalluQA.

**Strengths:**

1. The paper addresses a unique perspective on the issue, as many current works focus on hallucination detection and mitigation. The focus on automatically constructing high-quality hallucination datasets is crucial for addressing hallucinations in large language models. This is an important and valuable contribution to the field.
2. The structure of the paper is clear, and the writing is concise. The experimental section is detailed and thorough, enhancing the credibility of the results.

**Weaknesses:**

1. The rationale behind using the EM algorithm to solve this problem is not clearly articulated. What considerations led to this choice?
2. How does the paper ensure that the EM algorithm converges through iterations?

**Questions:**

1. The difficulty in automatically constructing high-quality hallucination datasets is not clearly explained. The relevant works in this area, the unresolved issues, and why these difficulties persist are not thoroughly discussed by the authors in the paper.
2. The rationale behind using the EM algorithm to solve this problem is not clearly articulated. What considerations led to this choice?
3. How does the paper ensure that the EM algorithm converges through iterations?

---

> ### Author Rebuttal · Authors · 2024-08-07
>
> Thanks for your thoughtful comments. Following are our responses to each individual comment (which are highlighted in italics).
>
> ### **Response to Weakness1 about contributions:**
>
> > *The description of the methodological contributions in the paper is not very clear. The EM algorithm and the self-consistency method mentioned in the article are existing works.*
>
> Our primary contribution is an iterative self-training framework that simultaneously and progressively scales up the hallucination annotation dataset and improves the accuracy of the hallucination annotator.
>
> Within this framework, the EM algorithm serves as our theoretical foundation. Additionally, we identified the need for a pipeline during the E-Step to produce more stable output. So we selected self-consistency as a method to achieve this stability.
>
> To the best of our knowledge, we are the first to present work on the automated construction of large-scale hallucination datasets.
> Moreover, the large-scale hallucination dataset and high-precision hallucination annotation model that we finally obtained can serve as the foundation for more research in the future, which we believe is very meaningful.
>
> ### **Response to Weakness2 and Question1 about the difficulty of automatic data construction:**
>
> > *The difficulty in automatically constructing high-quality hallucination datasets is not clearly explained. The relevant works in this area, the unresolved issues, and why these difficulties persist are not thoroughly discussed by the authors in the paper.*
>
> The difficulty of automatic hallucination data construction is the low performance of automatic annotators. The relevant works in this area, the unresolved issues, and why these difficulties persist were discussed in the Introduction (lines 32-35) and Related Work (lines 94-100).
>
> To solve these difficulties, we first proposed a progressively self-iterative labeling method that automatically scales up the dataset of fine-grained hallucination annotations, and we also proved the effectiveness of our method.
>
> ### **Response to Weakness3 and Question2 about the reason for using EM:**
>
> > *The rationale behind using the EM algorithm to solve this problem is not clearly articulated. What considerations led to this choice?*
>
> Our task can be formalized as optimizing two hidden variables, the hallucination annotator parameters and the data labels, as described in Section 3.2 (lines 138-140). We believe that the form of our task corresponds exactly to the EM algorithm [1].
>
> ### **Response to Weakness4 and Question3 about EM convergence:**
>
> > *How does the paper ensure that the EM algorithm converges through iterations?*
>
> EM is a convergent algorithm, as demonstrated in [2]. And, as illustrated by our experimental results, our approach has shown progressive improvement in annotation performance (Table 2) and generalization capabilities (Table 6) through iterations.
>
> [1] Dempster A P, Laird N M, Rubin D B. Maximum likelihood from incomplete data via the EM algorithm[J]. Journal of the royal statistical society: series B (methodological), 1977, 39(1): 1-22.
>
> [2] Wu C F J. On the convergence properties of the EM algorithm[J]. The Annals of statistics, 1983: 95-103.

---

### Official Review · Reviewer_4rBJ · 2024-07-12

**Soundness:** 3
**Presentation:** 3
**Contribution:** 3
**Rating:** 6
**Confidence:** 3

**Summary:**

The authors propose an innovative approach to tackle the persistent issue of hallucinations in large language models (LLMs) during long-form question-answering tasks. Current methods for detecting and mitigating these hallucinations are constrained by limited data and high labor costs. To address this, the paper introduces an iterative self-training framework that scales up the hallucination annotation dataset while simultaneously enhancing the accuracy of the annotators. Using the Expectation Maximization algorithm, the process involves multiple iterations where a pipeline annotates a growing dataset, and the improved annotator is used for the next cycle. This approach not only leads to a highly accurate annotator that surpasses GPT-4 but also effectively reduces hallucinations in LLM outputs. The results show significant improvements in key benchmarks, offering a scalable and efficient solution for managing LLM hallucinations.

**Strengths:**

1. One of the major strengths of this paper is its scalable framework. By automating the annotation process and iterating through larger datasets, the authors manage to overcome the usual limitations of manual data labeling, making the approach both cost-effective and efficient.

2. The application of the Expectation Maximization algorithm to improve annotator accuracy through iterative training is a standout feature. This method not only refines the annotations with each cycle but also ensures that the annotator becomes progressively more reliable, which is a clever and effective use of existing statistical techniques.

**Weaknesses:**

1. One of the big drawbacks of the EM algorithm is its sensitivity to initial conditions. If you don't start with the right parameters, the algorithm can easily get stuck in a local maximum instead of finding the best possible solution, which can be frustrating.

2.  The EM algorithm can be quite demanding in terms of computational resources. Each iteration involves a lot of number crunching, which means it can be slow and resource-intensive, especially when dealing with large datasets or complex models.

3. The authors could enhance their literature review by including several highly relevant papers on data annotation. The annotation task has been discussed in [1] [2] [3] [4] [5].

[1] https://arxiv.org/abs/2310.04668
[2] https://arxiv.org/abs/2303.15056
[3] https://arxiv.org/abs/2306.04349
[4] https://dl.acm.org/doi/pdf/10.1145/3613904.3642834
[5] https://dl.acm.org/doi/pdf/10.1145/3594536.3595161

**Questions:**

See weaknesses.

---

> ### Author Rebuttal · Authors · 2024-08-07
>
> Thank you for your constructive comment. Following are our responses to each individual comment (which are highlighted in italics).
>
> ### **Response to Weakness1 about initial condition:**
>
> > *One of the big drawbacks of the EM algorithm is its sensitivity to initial conditions. If you don't start with the right parameters, the algorithm can easily get stuck in a local maximum instead of finding the best possible solution, which can be frustrating.*
>
> We acknowledge the sensitivity of the EM algorithm to initial conditions.
>
> To address this problem, the first step of our framework was to train an annotator model with a high-quality, human-labeled hallucination dataset, ANAH [1]. We then obtained a model with annotation accuracy close to GPT4, and we used this high-accuracy model as a starting point for subsequent EM operations.
>
> Meanwhile, to ensure the stability of the convergence process, we use a progressive scaling strategy. Tables 2 and 4 show the effectiveness of this approach. In Table 2, the performance of the labeler improves continuously with data scaling, and in Table 4, the progressive approach is much more effective than the non-progressive approach.
>
> Although we cannot claim that we will eventually converge to a globally optimal solution, a flat convergence region would be nice according to the theoretical analysis [2, 3]. One measure of flatness is generalisability [4, 5], and Table 6 shows that our model has excellent generalisability. Thus, we believe that our method achieves a promising result.
>
> We will add a corresponding discussion in the next version.
>
> ### **Response to Weakness2 about computational resources:**
>
> > *The EM algorithm can be quite demanding in terms of computational resources. Each iteration involves a lot of number crunching, which means it can be slow and resource-intensive, especially when dealing with large datasets or complex models.*
>
> We agree that the iterative algorithm requires computational effort. However, it is important to consider our context where building a fine-grained hallucination annotation dataset requires prohibitively high costs and labor intensity.
>
> Using the "manual + GPT4-assisted" annotation model, as described in ANAH [1] (0.9 USD and 20 minutes per annotation), it would take 177,237 USD and 65,643 hours to reach the size of the dataset in our work.
>
> However, our method uses 32 A100 GPUs to iteratively train the 7B model. It took approximately 100 hours for inference and training. Based on the price of the computing platform Lambda (1.29 USD per GPU per hour), it only costs 4,128 USD.
>
> So we believe this is a better trade-off between computing resources and labour+API costs, which is acceptable.
>
> Moreover, the large-scale hallucination dataset and high-precision hallucination annotation model that we finally obtained can serve as the foundation for more research in the future, which we think is very meaningful.
>
> We will add a corresponding discussion in the next version.
>
> ### **Response to Weakness3 about relevant papers:**
>
> > *The authors could enhance their literature review by including several highly relevant papers on data annotation. The annotation task has been discussed in [6-10].*
>
> Thanks for your suggestion! We discuss the papers you mentioned below.
>
> [6] proposed a pipeline for annotating nodes on a graph without labels. [7, 8, 9] discussed the superiority of using GPT4 or ChatGPT as annotators. [10] introduces a self-supervised method using GPT for data annotation.
>
> Different from them, we introduce an iterative self-training framework that simultaneously and progressively scales up the hallucination annotation dataset and improves the accuracy of the hallucination annotator.
>
> We will add these relevant papers to our related work.
>
>
> [1] Ji Z, Gu Y, et al. ANAH: Analytical Annotation of Hallucinations in Large Language Models[J]. ACL, 2024.
>
> [2] Hochreiter S, Schmidhuber J. Flat minima[J]. Neural computation, 1997, 9(1): 1-42.
>
> [3] Hochreiter S, Schmidhuber J. Simplifying neural nets by discovering flat minima[J]. Advances in neural information processing systems, 1994, 7.
>
> [4] Keskar N S, Mudigere D, Nocedal J, et al. On large-batch training for deep learning: Generalization gap and sharp minima[J]. arXiv preprint arXiv:1609.04836, 2016.
>
> [5] Dziugaite G K, Roy D M. Computing nonvacuous generalization bounds for deep (stochastic) neural networks with many more parameters than training data[J]. arXiv preprint arXiv:1703.11008, 2017.
>
> [6] Chen Z, Mao H, Wen H, et al. Label-free node classification on graphs with large language models (llms)[J]. arXiv preprint arXiv:2310.04668, 2023.
>
> [7] Gilardi F, Alizadeh M, Kubli M. ChatGPT outperforms crowd workers for text-annotation tasks[J]. Proceedings of the National Academy of Sciences, 2023, 120(30): e2305016120.
>
> [8] He Z, Huang C Y, Ding C K C, et al. If in a Crowdsourced Data Annotation Pipeline, a GPT-4[C]//Proceedings of the CHI Conference on Human Factors in Computing Systems. 2024: 1-25.
>
> [9] Savelka J. Unlocking practical applications in legal domain: Evaluation of gpt for zero-shot semantic annotation of legal texts[C]//Proceedings of the Nineteenth International Conference on Artificial Intelligence and Law. 2023: 447-451.
>
> [10] Pei X, Li Y, Xu C. Gpt self-supervision for a better data annotator[J]. arXiv preprint arXiv:2306.04349, 2023.

---

> > ### Comment · Reviewer_4rBJ · 2024-08-13
> >
> > The authors have satisfactorily addressed most of my problems.  Most concerns has been addressed and some senarios may out of scope of this paper. I have raised my score. Once again, I want to express my gratitude for your hard work and commitment.

---

> > > ### Author Response · Authors · 2024-08-13
> > >
> > > Thank you for your response and for increasing the rating to 6 (Weak Accept). We are happy that our discussions on algorithm performance, computational resources, and related works are convincing. We will include these discussions in the final manuscript.

---

### Official Review · Reviewer_nXFU · 2024-07-16

**Soundness:** 4
**Presentation:** 4
**Contribution:** 4
**Rating:** 8
**Confidence:** 4

**Summary:**

This paper proposes an iterative self-training framework that simultaneously and progressively scales up the hallucination annotation dataset and improves the accuracy of the hallucination annotator. The framework is based on the expectation maximization algorithm, alternately annotating a scaled dataset and training a more accurate hallucination annotator on the dataset. A 7 billion model trained by this framework can surpass GPT-4 and obtains state-of-the-art hallucination detection results on HaluEval and HalluQA by zero-shot inference.

**Strengths:**

- The paper is well motivated by the fact that large language model hallucination significantly hinders applications but its annotation is difficult and very labor intensive.
- The solution based on self-training effectively and feasibly addresses the above challenge. (1) The paper defines a procedure of analytical hallucination annotation, that aligns with human cognitive processes. (2) Staged multi-dimensional data scaling, collecting synthetic data from more large language models and for more numbers of topics and questions, ensures the richness of the dataset. (3) Leveraging EM algorithm is suitable.
- Strong empirical results are shown for both in-domain hallucination detection and on existing benchmarks, HaluEval and HalluQA.
- Very well-written paper.

**Weaknesses:**

The RougeL and BertScore may not be the most capable metrics for evaluating generated texts.

**Questions:**

Is it possible to leverage large language model based evaluation for hallucination detection?

**Limitations:**

Appendix E describes the limitations.

---

> ### Author Rebuttal · Authors · 2024-08-07
>
> Thank you for your valuable feedback and for recognizing our efforts! Here are the answers to your question (which are highlighted in italics) regarding the metrics used to evaluate generated texts.
>
> ### **Response to Weakness and Question:**
>
> > *The RougeL and BertScore may not be the most capable metrics for evaluating generated texts. Is it possible to leverage large language model based evaluation for hallucination detection?*
>
> We use RougeL and BertScore as metrics because they are classical and popular metrics in NLG [1-3].
>
> In line with your suggestion, we additionally added an LLM-based evaluation metric. Specifically, we use FactScore [4] to assess the consistency of generated reference points with the source document.
>
> Below is Table 2 from our paper, which contains the newly added metric FactScore (column 3). The results of the FactScore indicates that the reliability of our model's generated reference points progressively improves and ultimately exceeds that of GPT4. This trend is consistent with F1 and ACC, reflecting the reliability of the FactScore.
>
> | Model          | F1    | ACC   | **FactScore** | RougeL | BertScore |
> |----------------|-------|-------|---------------|--------|-----------|
> | GPT-4          | 87.11 | 86.97 | **84.39**     | 86.32  | 96.21     |
> | ANAH-7B        | 78.69 | 79.92 | **80.60**     | 58.51  | 87.27     |
> | ANAH-20B       | 80.49 | 81.01 | **81.51**     | 58.82  | 88.44     |
> | ANAH-V2-Stage1 | 84.45 | 84.85 | **83.63**     | 60.10  | 88.43     |
> | ANAH-V2-Stage2 | 87.75 | 88.18 | **84.54**     | 67.28  | 90.80     |
> | ANAH-V2-Stage3 | 89.30 | 89.55 | **86.36**     | 69.44  | 91.43     |
>
> We will add a corresponding discussion and evaluation results in the next version.
>
> [1] Sai, Ananya B., Akash Kumar Mohankumar, and Mitesh M. Khapra. "A survey of evaluation metrics used for NLG systems." ACM Computing Surveys (CSUR) 55.2 (2022): 1-39.
>
> [2] Li, Junyi, et al. "HaluEval: A Large-Scale Hallucination Evaluation Benchmark for Large Language Models." EMNLP 2023.
>
> [3] Pagnoni, Artidoro, Vidhisha Balachandran, and Yulia Tsvetkov. "Understanding Factuality in Abstractive Summarization with FRANK: A Benchmark for Factuality Metrics." ACL 2021.
>
> [4] Min S, Krishna K, Lyu X, et al. Factscore: Fine-grained atomic evaluation of factual precision in long form text generation[J]. arXiv preprint arXiv:2305.14251, 2023.

---

> > ### Comment · Reviewer_nXFU · 2024-08-14
> >
> > Thank you for your response. My question was addressed and I keep the rating.

---

> > > ### Author Response · Authors · 2024-08-14
> > >
> > > Thank you for your response and for keeping the 8 (Strong Accept) rating. We are happy that our discussions and evaluation results about metrics have addressed your question. We will include these discussions in the final manuscript.

---

### Author Rebuttal · Authors · 2024-08-07

We sincerely appreciate the valuable feedback from the reviewers!

We are honored that our work can be reviewed as:

- The paper is "well motivated" and addresses the critical issue (R-5wgD).
- It provides a novel, effective and feasible solution (R-nXFU & 4rBJ & i3ot), and "obtains strong and thorough empirical results" (R-nXFU & 5wgD & i3ot).
- The paper  is "well written" (R-nXFU & 5wgD & i3ot) and well explained by figures and equations (R-i3ot).

For each question from all reviewers, we have provided a specific response in the relevant section below.

Any additional clarification and discussion suggested by the reviewers will be included in the revised version.

---

### Decision · Program_Chairs · 2024-09-25

**Decision:**

Accept (poster)

**Comment:**

This paper proposes an iterative self-training method to scale up hallucination annotation with better accuracy and uses the model to mitigate hallucination of LLMs. The proposed framework is scalable and efficient and shows strong performance on hallucination detection benchmarks. All the reviewers consistently acknowledge the contributions of this paper. Thus, I recommend acceptance.